# Cell Death: Mechanisms and Potential Targets in Breast Cancer Therapy

**DOI:** 10.3390/ijms25179703

**Published:** 2024-09-07

**Authors:** Jiangying Qian, Linna Zhao, Ling Xu, Jin Zhao, Yongxu Tang, Min Yu, Jie Lin, Lei Ding, Qinghua Cui

**Affiliations:** Lab of Biochemistry & Molecular Biology, School of Life Sciences, Yunnan University, Kunming 650091, China; qianjiangying1@stu.ynu.edu.cn (J.Q.); zhaolinna@stu.ynu.edu.cn (L.Z.); xuling_gyay@stu.ynu.edu.cn (L.X.); zhaojin_cujn@stu.ynu.edu.cn (J.Z.); tangyongxu@stu.ynu.edu.cn (Y.T.); yumin@ynu.edu.cn (M.Y.); linjie@ynu.edu.cn (J.L.)

**Keywords:** breast cancer, autophagy, apoptosis, pyroptosis, necroptosis, ferroptosis, cuproptosis

## Abstract

Breast cancer (BC) has become the most life-threatening cancer to women worldwide, with multiple subtypes, poor prognosis, and rising mortality. The molecular heterogeneity of BC limits the efficacy and represents challenges for existing therapies, mainly due to the unpredictable clinical response, the reason for which probably lies in the interactions and alterations of diverse cell death pathways. However, most studies and drugs have focused on a single type of cell death, while the therapeutic opportunities related to other cell death pathways are often neglected. Therefore, it is critical to identify the predominant type of cell death, the transition to different cell death patterns during treatment, and the underlying regulatory mechanisms in BC. In this review, we summarize the characteristics of various forms of cell death, including PANoptosis (pyroptosis, apoptosis, necroptosis), autophagy, ferroptosis, and cuproptosis, and discuss their triggers and signaling cascades in BC, which may provide a reference for future pathogenesis research and allow for the development of novel targeted therapeutics in BC.

## 1. Introduction

Breast cancer (BC) was defined as a Global Common Cancer by the International Agency for Research on Cancer (IARC) in 2020. Approximately 31% of new cases and 15% of deaths are due to BC each year, making it the second leading cause of death among women [1]. A study in 2023 showed that about 2.44 million BC patients died in 81 countries or regions, which represented a reduction of 3.2–11.6% over the past 20 years due to earlier diagnosis [2].

BC can be classified as HER2-rich type (hormone receptor (HR), estrogen receptor (ER), progesterone receptor (PR)^−^/HER2^+^), basal-like type (triple-negative breast cancer, TNBC, ER/PR/HER2^−^), normal mammary type [3,4], and luminal type, including luminal A (HR^+^/HER2^−^) and luminal B (triple positive) [4,5]. Endocrine therapy and chemotherapy are commonly used to treat BC [3]. HER2 was used recently as a target for BC treatment, but this was not effective in TNBC due to a lack of target receptors.

Cell death has been a focus of biological and cancer treatment studies. Understanding the signaling pathways and regulatory mechanisms in cell death could provide potential therapeutic targets for BC, as well as other types of cancer [6,7,8,9,10]. PANoptosis has emerged as a complicated cell death based on the crosstalk of pyroptosis, apoptosis, and necroptosis [11,12,13], dependent or independent of autophagy, which is involved in BC treatment or drug resistance. Ferroptosis and cuproptosis have been identified as novel cell death patterns, induced and mediated by the metal ions Fe and Cu, respectively. Ferroptosis is driven by lipid peroxide [14], while cuproptosis is caused by the accumulation of copper ions, leading to cell poisoning [15]. The profiles of each type of cell death are illustrated in Figure 1.

## 2. Autophagy

### 2.1. Autophagy: A PCD Mediated by Lysosomes

The concept of autophagy was first proposed by Christian de Duve, stemming from his discovery that the intraperitoneal injection of rats with glucagon induced the formation of autophagic vacuoles and affected the lysosomes [16]. Initially, researchers observed the degradation of the cell structure and the emergence of lysosomes in rat liver cells and then observed autophagy in *Saccharomyces cerevisiae* [17]. After that, autophagy-related genes were identified [18]. Autophagy is a highly conserved cellular physiological and pathological process widely found in eukaryotes and is a tightly regulated programmed cell death (PCD). It plays an important role in maintaining cellular homeostasis and removing damaged or senescent organelles, bacteria, viruses, and other exogenous substances.

Autophagy can be categorized as macroautophagy, chaperone-mediated autophagy (CMA), and microautophagy. Macroautophagy is also generally referred to as autophagy. It is the process by which autophagosomes with double-membrane structures fuse with lysosomes and degrade their contents. CMA is a process in which the substrate proteins with a Lys-Phe-Glu-Arg-Gln (KFERQ) motif are recognized and transported to lysosomes for degradation. After binding to chaperone heat shock cognate protein 70 (HSC70), the substrate proteins are transported to lysosomes and then recognized and internalized by lysosomal-associated membrane protein 2A (LAMP2A) for degradation by lysosomes [19]. Microautophagy is a process of lysosomal or endosomal membrane self-trapping, encapsulating cytoplasmic proteins into lysosomes and degrading them by lysosomal proteases [20]. Autophagy can also be categorized as non-specific or specific autophagy. Non-specific autophagy is the process by which autophagosomes randomly engulf cytoplasmic components and break them down into cellular debris to provide energy under starvation conditions, while specific autophagy is the process by which the autophagosomes selectively phagocytose substances, such as damaged or senescent organelles or other macromolecular substances [21]. Depending on the selectivity of specific autophagy for substrates, autophagy can be categorized as mitochondrial autophagy, peroxisomal autophagy, chloroplast autophagy, ribosomal autophagy, ER-phagy, lysophagy, lipophagy, and so on.

The process of autophagy has five stages: initiation, nucleation, vesicle membrane extension, vesicle fusion, and degradation of intravesicular material. In the initiation phase, when mTORC is inhibited under starvation or cell energy deficit conditions, FIP200 and autophagy-related gene 13 (ATG13) are phosphorylated by phosphorylated UNC51-like kinase-1 (ULK1), thereby activating the ULK complex. The activation of AMPK can also cause the ULK complex to take the excitation form. The ULK complex is composed of FIP200, ATG13, and ULK1/2 [22]. At the nucleation stage, the ULK complex is activated and binds to the phosphatidylinositol 3-kinase (PI3K) complex to produce phosphatidylinositol 3-phosphate (PIP3), which can recruit more autophagy-associated proteins after localization in autophagosome precursors, promoting autophagosome formation and membrane extension. The PI3K complex consists of Beclin-1, VPS34, and ATG14. During the vesicle membrane extension stage, ATG5, ATG12, and ATG16 combine to form the E3 ubiquitin ligase. Microtubule-associated protein 1 light chain 3 (LC3) is the homolog of the yeast autophagy gene ATG8 in humans and is essential for membrane extension [23]. LC3 is converted to soluble LC3-I in the presence of ATG4. LC3-I is then transformed into lipoylated LC3-II in the presence of E1, E2, and E3. LC3-II is bound to phosphatidylethanolamine (PE) and inserted into the autophagosome membrane, where it helps extend the membrane [24]. During membrane extension to form autophagosomes, specific receptor proteins recognize and bind molecules with specific degradation signals to form complexes that are encapsulated into autophagosomes and thus degraded by lysosomes [25,26].

### 2.2. Autophagy-Related Genes Dysregulation and Therapy in BC

Autophagy has a dual role in tumor progression and treatment. On the one hand, it promotes tumor survival by providing nutrients and energy for the rapid proliferation of cancer cells through the degradation of organelles and macromolecules. Meanwhile, autophagic activity is enhanced after tumor radiotherapy, aiding in the elimination of damaged organelles and macromolecules caused by the treatment [27,28]. On the other hand, autophagy inhibits tumor survival. During the replication crisis, autophagy helps clear cancerous and mutated substances within cells, preventing normal cells from turning into cancer cells. The replication crisis refers to cell death caused by the shortening of telomeres, which serves as the last barrier to eliminating precancerous cells from transforming into cancer cells [29]. Few precancerous cells can escape the replication crisis; thus, it can act as an effective tumor suppressor. Autophagy has a crucial role during the replication crisis in activating the death of precancerous cells. When autophagy is inhibited, precancerous cells are more likely to bypass the replication crisis, and this can also promote cell proliferation and the accumulation of genomic instability in precancerous cells. The occurrence of cancer is often accompanied by the loss of autophagy function [30].

Currently, BC is treated via chemotherapy, endocrine therapy, targeted therapy, radiation therapy, and immunotherapy [31]. Autophagy plays an important role in disease development in melanoma, colorectal cancer, pancreatic cancer, and neurodegenerative diseases [32]. Altered autophagy affects the development and treatment of BC in many ways.

The Beclin-1 plays a very important role in autophagy, hence, the Beclin-1 can be used as a potential therapeutic target for BC. Beclin-1 has a dual role in BC development, both inhibiting tumor growth and promoting tumor survival. In most BC carcinogenesis, the expression level of Beclin-1 is reduced [33]. In addition, Beclin-1-deficient heterozygous mice showed an increased risk of developing breast, ovarian, and prostate cancers [34]. Conversely, in Beclin-1 overexpressing mice, cancer risk is reduced, particularly with HER2 overexpression BC, due to increased autophagy [35]. However, after the knockout of Beclin-1, the proliferation of BC cells is enhanced, which promotes the progression of BC [35], although some studies have shown that the expression of Beclin-1 increases autophagy activity, which in turn inhibits BC growth. However, some research has also illustrated that a high expression level of Beclin-1 in vitro increases the level of cellular autophagy, improving the survival of BC cells under the conditions of starvation and hypoxia [34]. In particular, in HER2-positive BC and TNBC, overexpression of Beclin-1 is associated with more aggressive properties [36].

MicroRNAs (miRNAs) are small non-coding RNA molecules composed of approximately 15 to 25 nucleotides that can regulate the expression of multiple autophagy genes. Dysregulated miRNA expression has been found in BC [37], suggesting miRNAs may provide new options for BC therapy. Ruiz et al. identified 18 miRNAs that inhibited autophagic activity by downregulating the expression of ULK1 and inhibiting the phosphorylation of Beclin-1 in the BC cell line MDA-MB-231. These 18 miRNAs can also inhibit autophagy by inhibiting the MAPK/JNK signaling pathway to dephosphorylate Bcl-2, which thus binds to Beclin-1 more easily [38]. MiR-489 reduces the expression of ULK1 in various BC cell lines, thereby decreasing autophagy activity and inhibiting tumor growth. MiR-489 can also inhibit autophagy to increase the sensitivity of BC cells to adriamycin, that is, to reduce their resistance to adriamycin, thereby effectively treating BC [39]. MiR-27a can inhibit autophagy by blocking substrate delivery to the autophagosome by decreasing the level of LC3-II and SQSTM1. It can also increase drug resistance in BC cells in vitro [40]. MiR-20a negatively regulates the levels of Beclin-1, ATG16L1, and SQSTM1, and it promotes the occurrence of BC by reducing autophagy activity [41].

Long non-coding RNAs (lncRNAs) are long RNA molecules that exceed 200 base pairs in length, with abnormal expression observed in BC. lncRNAs can directly regulate autophagy, and their overexpression in BC affects cancer metastasis and drug resistance [42]. Consequently, lncRNAs can serve as therapeutic targets to influence the progression of BC. The first lncRNA associated with cancer progression was HOX antisense intergenic RNA (HOTAIR) [43], which significantly up-regulates target RNA molecules expressed in BC [44]. Li et al. found that the knockdown of HOTAIR inhibited the PI3K/AKT/mTOR pathway, leading to a reduction in BC resistance to adriamycin [45]. Another lncRNA associated with BC is NAMPT-AS, which is located in the nicotinamide phosphoribosyltransferase (NAMPT) promoter region and is overexpressed in TNBC. NAMPT-AS overexpression enhances the viability, proliferative capacity, and oncogenic properties of TNBC cells, and is associated with a poor prognosis for patients. NAMPT-AS regulates TNBC progression by modulating the transcriptional activity of NAMPT. After downregulation of NAMPT expression, there is an increase in the expression of ATG5, ATG7, and Beclin-1. This enhances autophagy activity, thereby inhibiting cell survival and invasiveness [46]. Additionally, another lncRNA, differentiation antagonizing non-protein-coding RNA (DANCR), is overexpressed in BC, and knocking it down can lead to decreased cell proliferation. Both LC3B and ATG5 are elevated, leading to enhanced autophagy, which inhibits the malignant progression of BC [47].

## 3. PANoptosis: A Cross-Regulated Mode of Cell Death

PANoptosis, first named in 2019, is a newly described form of cell death that combines three modes of cell death—apoptosis, pyroptosis, and necroptosis—to form a complementary circuit that compensates for cell death escape [48]. Before the discovery of PANoptosis, researchers did not know how these cell processes could be connected by a complex that acts as an amplifying cell death mechanism. This discovery suggests that PANoptosis can regulate the overall cell death process, making it a powerful means of inhibiting cancer. However, it is important to note that while PANoptosis can inhibit cancer, mutations in genes lead to cancer progression.

Apoptosis, pyroptosis, and necroptosis have a lot of overlap in mechanisms. For example, the binding of death receptors (DRs) and death ligands (DLs) can activate both apoptosis and necroptosis. When tumor necrosis factor receptor 1 (TNFR1) binds to its ligand, TNF-related apoptosis-inducing ligand (TRAIL), it will recruit a complex I composed of tumor necrosis factor receptor-associated death domain protein (TRADD), receptor-interacting protein 1 (RIP1), and E3 ubiquitin ligases. After that, RIP1 is deubiquitinated, prompting the formation of complex IIa or IIb. The complex IIa can activate caspase-8 and induce extrinsic apoptosis. When caspase-8 is inhibited, the complex IIb will form and activate necroptosis [11]. Furthermore, the caspase family can be found in both apoptosis and pyroptosis. The caspase-1/4/5/11 can cleave GSDMD and induce pyroptosis [49,50,51,52], while the caspase-8/9 can cleave caspase-3 and induce apoptosis [53,54]. Meanwhile, caspase-3 can also cleave GSDME and induce pyroptosis, which depends on the activation of GSDME [55]. The ROS can also affect apoptosis, pyroptosis, and necroptosis. ROS can promote the nuclear localization and transcriptional activation of NF-κB [56]. NF-κB can promote the expression of Bcl-X_L_, which can inhibit apoptosis [57]. ROS can activate Nod-like Receptor Protein 3 (NLRP3) inflammasome, promote caspase-1, and induce pyroptosis [58]. ROS promotes necrosome formation and necroptosis by affecting RIP1 phosphorylation [59].

### 3.1. Apoptosis

#### 3.1.1. Apoptosis: A PCD Induced by Caspases

Apoptosis, as a form of cell death with the same long history as autophagy, was first defined in 1972 by Kerr et al. [60]. It is an energy-dependent cell death in which chromatin is dispersed and condensed, cells shrink, DNA is cut, and the nucleus divides into vesicles under the control of apoptosomes, which are then phagocytosed and decomposed by macrophages or neighboring cells [60,61]. Apoptosis is mainly mediated by caspase family proteins and can be divided into two pathways: intrinsic apoptosis and extrinsic apoptosis [62]. Caspase-3, the executioner of apoptosis, can cause cell death by cleaving α-tubulin [63], Mcl-1 [64], Actin [63], Fodrin [65], Lamin [66], PARP [67], ICAD [68], and other substrate proteins.

The intrinsic pathway of apoptosis is also called the mitochondrial pathway, which depends on caspase-9 and Bcl-2 family proteins. Bcl-2 family proteins can be divided into two opposite types: one is anti-apoptosis and involves the Bcl-homology (BH) 1/2/3 domain, including Bcl-2, Bcl-xl, Bcl-w, et al., and the other is pro-apoptosis and involves the BH3 domain only, including Bax, Bak, Bad, Bok, et al. [69]. Regulation of the mitochondrial pathway is inseparable from the BH domain. For Bax, Bcl-2 with BH1/2/3 can form a heterodimer with Bax with BH3 only, thus inhibiting apoptosis [70]. When cells are stimulated by apoptotic signals, the expression of Bax increases, enabling it to avoid binding to the Bcl-2 protein. It oligomerizes and locates to the mitochondrial outer membrane and forms channels [61], or binds to the voltage-dependent anion channel (VDAC) protein to form larger channels on the mitochondrial outer membrane [71,72]. Then, cytochrome C (CytC), via the mitochondrial membrane gap, can be released into the cytoplasm [73]. Bak is localized to the mitochondrial outer membrane. After the apoptotic signal is generated, dormant Bak is activated by the BH3 domain of the BH3 interacting domain death agonist (BID), after which the activated Bak can recruit and activate floating dormant Bak to form a cascade reaction. These activated Bak oligomerize to form holes in the mitochondrial outer membrane, leading to mitochondrial outer membrane permeabilization (MOMP) [74]. When CytC is released into the cytoplasm, it can bind to apoptotic protease activating factor-1 (APAF1) monomers to change its conformation, exposing the WD domain, oligomerizing several APAF1 to assemble apoptosome by binding dATP [54], and then exposing the caspase activation and recruitment domain (CARD) of APAF1 to recruit and activate caspase-9, catalyze caspase-3 maturation, and induce apoptosis. In addition, as an activating transcription factor, p53 can induce mitochondrial pathway apoptosis by promoting *PUMA* (a p53 up-regulated modulator of apoptosis) expression, specifically of PUMA-α and PUMA-β, which contain the BH3 domain [75].

The extrinsic pathway of apoptosis, which is also called the DR-mediated pathway, depends on NK cells binding to DRs and activating caspase-8 to induce apoptosis. The DRs include Fas receptors, DR4/DR5, tumor necrosis factor receptors (TNF-R), and TNF-related apoptosis-inducing ligand receptors (TRAIL-R), which are all present on the surface of cells [76]. After extracellular DL (TNF) binds to DR (TNF-R) on the cell membrane, the adaptor protein TRADD is gathered and recruits FADD, then monomeric pro-caspase-8 is recruited to form the death-inducing signaling complex (DISC) through the DED domain of FADD. Pro-caspase-8 interacts with DISC and promotes dimerization of pro-caspase-8 and is activated by caspase-8 [11,53]. The activation of apoptosis via caspase-8 is influenced by two types of cells. The “type II” cell induction of apoptosis must occur by activating mitochondria and the “type I” cell induction does not [53]. In “type I” cells, caspase-8 can activate caspase-3 directly, while in “type II” cells, the mitochondria should release inhibitors such as Smac/DIABLO [53] and HtrA2/Omi [77] to inhibit the inhibitor of apoptosis proteins (IAPs), or IAPs will inhibit caspase-8 to activate other caspases and prevent apoptosis. The whole process of the intrinsic and extrinsic apoptosis pathways is illustrated in Figure 2.

#### 3.1.2. Apoptosis-Related Genes Dysregulation and Therapy in BC

Apoptosis plays an important role in the occurrence and development of BC. Studies have shown that apoptosis is involved in various processes of normal breast development, and the occurrence of BC is mainly due to the reduction of apoptosis, which breaks the balance between cell proliferation and cell death during breast development [78]. There are a lot of abnormal expressions of apoptosis-related genes in the process of BC.

The Bcl-2 family proteins involved in apoptosis are always expressed abnormally in BC. In humans, Bcl-2 is expressed in about 80% of BC cases [78] and is overexpressed in about 75% of BC cases [79]. Therefore, the expression of Bcl-2 with apoptosis inhibition can be used as a criterion for judging the prognosis of BC. Alipour et al. designed a nanocarrier called MiRGD, which can bind to DNA and accurately deliver it to BC cells, promoting BC cell apoptosis by silencing Bcl-2 [80]. In addition, it is possible to construct a BH3 domain-like structure to inhibit Bcl-2 through domain interaction. Bcl-2 can be inhibited by the BH3 mimetic ABT-737 or 199 (venetoclax) combined with tamoxifen, which can more effectively inhibit BC cell growth and reduce drug resistance [81]. Bcl-2 can also be inhibited by promoting the expression of Bcl-2 inhibition proteins. For example, the expression of Bim, Bax, or Bak can be promoted by oleandrin and quercetin, which promote the apoptosis of BC cells by inhibiting the expression of Bcl-2 to inhibit BC growth [82,83].

Further, direct promotion of caspases can also promote apoptosis in BC cells. Caspase-3 can be highly expressed and activated by promoting MOMP, and then inhibiting anti-apoptotic proteins and activating intrinsic apoptosis, thereby inducing apoptosis and inhibiting BC cell proliferation and migration [84]. The cleavage of pro-caspase-9/3/7 and mitochondrial CytC releasing can be promoted by Diallyl trisulfide in BC cells, leading to PARP proteolysis, and inducing both intrinsic and extrinsic apoptosis [85].

Various signaling molecules that induce apoptosis can also aid BC treatment. In human BC cells, apoptosis induced by IL-4 may be related to phosphorylation of IRS-1 and STAT6. STAT6 can dimerize and translocate to the nucleus after phosphorylation, activating promoters of specific genes [86]. Furthermore, IL-4 can promote apoptosis by inhibiting ERK, thereby promoting caspase-8 activation [87]. Interferon γ (IFNγ) can promote the expression of FasL and control apoptosis of BC cells via the Fas:Fc fusion protein transmitting signal [88]. Moreover, controlling HtrA2/Omi release can reduce TRAIL resistance and promote BC cell apoptosis [89]. TRAIL and FasL signaling in BC can be disrupted by miR-519a-3p and induce resistance to apoptosis, because miR-519a-3p can reduce the expression level of TRAIL-R2 and caspase-7/8, leading to the death escape of cancer cells [90]. Moreover, inhibiting cyclin D1 and mTOR, up-regulating PPARγ expression [91], interfering with the PI3K/AKT [92] or NF-κB [93] signaling pathway by cannabidiol, can induce apoptosis, promote cell differentiation, and prolong the cancer cell proliferation in BC. Inhibiting topoisomerases to interfere with DNA replication via chemotherapeutic drugs such as quinacrine and doxorubicin (DOX, or adriamycin) can cause cell cycle arrest and DNA damage, and induce p53 expression or interfere with DNA damage repair pathways [94,95], thereby leading to apoptosis in BC cells. The pro-apoptotic tumor suppressor protein Par-4 can promote tumor regression and reduce BC recurrence by stably expressing p53, stimulating multinucleation, inhibiting proliferation, and promoting apoptosis in BC cells [96].

### 3.2. Pyroptosis

#### 3.2.1. Pyroptosis: A PCD Induced by the GSDMs

Gasdermin family proteins (GSDMs) are mainly related to diseases such as hearing impairment [97], alopecia [98], childhood asthma [99], neurodegenerative pathologies [100], breast cancer [101], melanoma, and gastric cancer [102,103]. Six categories of GSDMs have been identified, including GSDMA, GSDMB, GSDMC, GSDMD, GSDME/DFNA5, and PJVK/DFNB59 [104,105]. The proteins, which are mainly expressed in the gastrointestinal tract and skin [104,106,107], were found to have new functions of causing inflammatory cell death via pyroptosis in 2015 [50,108,109]. Pyroptosis was first described in 2001 [110,111], although at that time the exact mechanism was unknown and it was only understood as a new form of cell death caused by caspase proteins that could cause inflammation. The basic mechanism of pyroptosis is now known. It is a form of PCD mediated by GSDMs that release interleukins to trigger inflammation. Pyroptosis induced by GSDMDs can be divided into the classical pathway and the non-classical pathway.

The classical pyroptosis pathway is mediated by caspase-1 and is dependent on GSDMD [49]. Intracellular GSDMs typically consist of a C-terminal repressor domain (RD) and an N-terminal pores-forming domain (PFD), also known as CT and NT [102,112]. Only isolated NT fragments can bind to membrane lipids, causing perforation. Usually, intracellular GSDMs are in a self-inhibitory state (except PTVK) [108,113]. However, under bacterial or viral invasion (known as pathogen-associated molecular patterns or PAMPs), the intracellular pattern recognition receptor Nod-like Receptors (NLRs) recognize these PAMPs and assist NLRP3 in binding to the adaptor protein ASC (apoptosis-associated speck-like protein, containing a CARD). The CARD domain of the ASC will recruit pro-caspase-1 and assemble to form the NLRP3 inflammasome [114,115], activating caspase-1 and cleaving GSDMD to release its cytotoxic NT fragment [108], causing cell membrane perforation and cell death. At the same time, the cell contents are released outside of the cell, which triggers an inflammatory response. Activated caspase-1 also cleaves pro-IL-1β and pro-IL-18 [116,117], activating IL-1β and IL-18 and releasing them extracellularly [114,118], leading to the aggregation of inflammatory cells and increasing the inflammatory response.

The non-classical pathway of pyroptosis is mediated by caspase-4/5/11 [50,51,52]. The main difference between the non-classical pathway and the classical pathway is that, in the non-classical pathway, there is no need for an ASC adaptor [51,119] to assist in caspase-4/5/11 monomer recruitment. Additionally, in the non-classical pathway, caspase-4/5/11 is unable to cut pro-IL-1β and pro-IL-18, which limits its ability to amplify the inflammatory response caused by pyroptosis. However, in non-immune cells, caspase-4 can cleave pro-IL-18 after conformational changes after cleavage [120]. Caspase-11 can also promote the classical pyroptosis pathway and IL-1β maturation and secretion by assisting caspase-1 processing [121]. The CARD domain of caspase-4/5/11 [52] can recognize and bind to lipid A of the pathogen lipopolysaccharides (LPS) [51,52,121]. Under the stimulation of LPS, caspase-4/5/11 is activated by oligomerization [52] and cleaves GSDMD to release NT fragments [51], inducing pyroptosis. Both the classical and non-classical pathways of pyroptosis are illustrated in Figure 3.

In addition to GSDMD, other proteins of the GSDM family can also cause pyroptosis. Pyroptosis caused by GSDME is also called the apoptotic caspase pathway because caspase-3, which is responsible for cleaving GSDME, is also used to activate apoptosis. However, in the case of elevated GSDME expression, apoptosis is converted to pyroptosis due to the cleavage of GSDME by caspase-3, releasing NT fragments with pore-forming activity [124,125]. GSDMB can be cleaved by granzyme A to release GSDMB-NT and cause pyroptosis [126]. There is a positive feedback pathway of pyroptosis, in which the immune microenvironment, activated by pyroptosis, CD8+ T cells, and NK cells, can further promote pyroptosis by cleaving GSDME through granzyme B [12]. GSDMC can be cleaved by caspase-6, caspase-8 [127,128], and TNF-α, thus causing pyroptosis.

#### 3.2.2. Pyroptosis-Related Genes Dysregulation and Therapy in BC

Pyroptosis has dual effects on cancer because it is accompanied by the release of inflammatory factors IL-1β and IL-18, and inflammatory factors may have some adverse effects on the tumor microenvironment (TME) during BC development. IL-1β is always overexpressed in BC cells compared to normal cells [129,130,131]. IL-1β and IL-18, as inflammatory factors, can promote immune cell accumulation. IL-1β can also promote the growth of blood vessels in tumors [132], including in BC. NLRP3 has been implicated in tumor myeloid cell infiltration due to its ability to promote IL-1β maturation through activation of caspase-1 [133]. As well as IL-1β overexpression in BC cells, NLRP3 has also been found to be overexpressed in BC cells [134]. IL-1β also can promote BC cell migration and invasion by activating ERK1/2 [129], which can activate the MAPK signaling pathway.

GSDME expression levels are usually lower in BC cells than in normal cells [125]. The expression of GSDME was increased through DNA demethylation, thus inducing pyroptosis, increasing chemosensitivity, and reversing chemotherapy resistance to paclitaxel [135]. Activating caspase-3 can also promote the apoptotic caspase pathway of pyroptosis in BC cells, effectively inhibiting tumor metastasis [136]. ROS plays an important role in the DOX-treated BC cells, which can induce caspase-3/GSDME-mediated pyroptosis through the ROS/JNK pathway [137]. STAT3 phosphorylation can promote mitochondrial ROS production, induce pyroptosis by promoting the caspase-3/PARP/GSDME axis, and inhibit tumor growth and lung metastasis in TNBC [138].

The expression of GSDMB, which has six main protein isoforms [124], is up-regulated by about 65% in HER2 BC cells [139,140], especially GSDMB-2. The high expression of GSDMB-2 can induce tumor invasion and assist metastasis [101,141]. Other research showed that only GSDMB-3/4 has a stable pore formation structure β9-β11 hairpin, and the cavity formed by α3-β6-β9 can induce pyroptosis, making it beneficial in BC therapy. Other GSDMB isoforms can inhibit GSDMB-3/4-mediated pyroptosis [126]. In the mammary gland, GSDMB-2 promotes BC by assisting HER2 [142]. Aiming for anti-GSDMB action, Ángela et al. designed a drug called AbGB-NC, which is encased in hyaluronic acid nanocapsules (NC). This drug successfully suppressed most cancer metastasis in a mouse HER2 BC model, reducing the aggressiveness of HER2 BC [139].

GSDMC is highly expressed in BC [143,144], producing dual effects similar to those of GSDMB. GSDMC can induce pyroptosis of BC cells [127,128,143]. In BC cells, PD-L1 can transform TNF-α-induced apoptosis into pyroptosis by interacting with p-Stat3 to enhance the transcription of GSDMC [128]. Meanwhile, high levels of GSDMC can increase the sensitivity of cancer cells to poly (ADP-ribose) polymerase inhibitors (PARPi). PARPi can promote caspase-8/GSDMC-mediated pyroptosis, while GSDMC can promote cancer cell sensitivity to chemotherapy drugs via activating caspase-6/8 [127]. In TNBC, about 38.5% of cancer patients’ tumor tissues show high expression of GSDMC [127], indicating that PARPi has potential in TNBC treatment. In addition, Sun et al. showed that the LINC00511/hsa-miR-573/GSDMC axis could also increase GSDMC expression, but this up-regulates associated with a poor prognosis and tumor immune infiltration of BC [143]. However, further research is needed to understand why this association occurs.

### 3.3. Necroptosis

#### 3.3.1. Necroptosis: A PCD Induced by RIP and MLKL

Necroptosis is a form of cell death that occurs when apoptosis is inhibited. It does not rely on cysteine caspase, but instead involves receptor-interacting kinase 1 (RIPK1), receptor-interacting kinase 3 (RIPK3), and mixed-spectrum kinase structural domain-like (MLKL) [145]. Necroptosis can be triggered by a variety of stimuli and is most often induced by TNF [146]. After necroptosis is triggered by TNF-α, RIPK1 phosphorylates RIPK3, which forms the necrosome RIPK1/RIPK3 complex, and the activated RIPK3, in turn, phosphorylates and activates MLKL, which increases the permeability of the cellular membrane, causing the cell to rupture and release its contents, resulting in necroptosis [13].

#### 3.3.2. Necroptosis and Breast Cancer Therapy

Necroptosis plays an important role in BC development, so targeting necrotic-related proteins may be helpful for BC treatment. The expression of RIP3 in cancer tissues is significantly lower than in normal breast tissues for most BC patients [147]. This is because the RIP3 transcriptional site is methylated, leading to the repression of RIP3. As a result, downstream MLKL is not activated by its phosphorylation, and necroptosis is inhibited. Additionally, RIP3 repression contributes to greater drug resistance and promotes the survival of BC cells. Morgan et al. suggested that BC could be treated by adding a hypomethylating drug to induce RIP3 expression [147]. Jiao et al. observed a significant increase in MLKL expression and detected phosphorylation of MLKL in the necrotic regions of solid BC tumors. When MLKL was knocked down, there was less tumor cell death and fewer necrotic regions in the tumors. The tumors lost their ability to metastasize to the lungs in mice. This suggests that the formation of necrotic regions of tumors is strongly associated with necroptosis [148]. Stoll et al. found that MLKL and RIP3 promoted the expression of IFNα and IFNγ-related genes and enhanced anticancer immunity in a mouse model of BC [149].

Various drugs, such as cystine depletion [150], quercetin [151], Smac mimetic LCL161 [150], and non-benzoquinone analogs of menthamycin DHQ3 [152], have been demonstrated to promote BC cell death by inducing the onset of necroptosis. Many researchers are also exploring more diverse strategic approaches to induce necroptosis to treat BC, such as the addition of hypomethylating agents to induce the expression of RIP3 [147] and metal analogs to induce necroptosis to overcome drug resistance in BC [153].

## 4. Cell Death Modulated by Metal Ions in the Microenvironment

The cellular microenvironment can significantly impact the lifespan of cells, particularly the levels of metal ions present. Numerous metal ions serve as active centers for enzymes, allowing them to carry out their functions effectively. However, an excess of metal ions can be toxic to cells. The mechanisms of cell death regulation that have been researched and organized are ferroptosis and cuproptosis.

### 4.1. Ferroptosis

#### 4.1.1. Ferroptosis: A PCD Induced by Fe(II) and ROS

Iron ions play an important role in organisms. An iron ion is the metal active center of many kinds of proteins. As a vital protein in metabolism, the iron and sulfur (Fe-S) cluster participates in the electron transport chain, and it is a coenzyme factor of CytC and various enzymes. Iron ions are also an important component of heme and a carrier of oxygen.

Ferroptosis is a form of cell death in which ROS and lipid peroxides are produced by iron metabolism and accumulate to produce lethal toxicity due to failure to metabolize smoothly. ROS is an important factor leading to lipid peroxide production in cells and has strong oxidation like Fe(II). Ferroptosis is mainly regulated by GSH (glutathione) and Glutathione Peroxidase 4 (GPX4) [154,155], an enzyme capable of degrading lipid peroxides in cells, and the CoQH2 system, which depends on FSP1 (CoQ oxidoreductase) [156,157]. Since its first discovery in 2012 [14], ferroptosis has been identified in various diseases, especially in cancer treatment. When ferroptosis occurs, there are distinctive changes in mitochondrial morphology, including cristae reduction, outer membrane rupture, an increase in intracellular lipid ROS levels, and subsequent cell membrane rupture leading to cell death [14]. At the same time, biochemical indicators of iron ions are significantly increased [158]. The key to inducing ferroptosis lies in the production and accumulation of lipid peroxides with lethal toxicity. It mainly involves polyunsaturated fatty acid phospholipid hydroperoxide (PUFA-PL-OOH), while Fe (II), ALOXs/PEBP, POR, O_2_, and ROS can transform PUFA-PL into PUFA-PL-OOH [155], resulting in cell death.

The accumulation of PUFA-PL-OOH depends on the activities of GPX4 and FSP1. GPX4 can reduce PUFA-PL-OOH to non-toxic PUFA-PL-OH [155], while CoQH_2_, produced by FSP1, can capture lipophilic free radicals and inhibit the production of PUFA-PL-OO∙ [156,157]. If the activity of GPX4 and FSP1 is inhibited, intracellular lipid peroxides accumulate and diffuse, as the enzymes that can degrade them and CoQH_2_, which can prevent their transport, are lost [154,156,157]. The subsequent series of reactions after iron and hydrogen peroxide to generate hydroxyl radicals are collectively referred to as the Fenton reaction [159]. In the presence of Fe^2+^, the intracellular Fenton reaction will continuously generate ROS [160,161], further causing lipid peroxidation. In this way, a large amount of lipid peroxide accumulates in the cell, destroying the original enzyme system so that the cell is poisoned and degraded.

Ferroptosis is regulated in many ways. Firstly, the cystine import system SystemX_C_ ^−^ (cystine–glutamate transporter receptor) upstream of GPX4 is inhibited, and SystemX_C_^−^ is regulated by inhibitors such as Imidazole Ketone Erastin (IKE), Sulfasalazine, and Sorafenib [162]. SystemX_C_^−^ consists of a dimer of SLC7A11 and SLC3A2 located on the cell membrane, whose main function is to replace intracellular glutamate with extracellular cystine [163,164,165], allowing cystine to enter the cell. As the cofactor of GPX4, GSH is synthesized from glutamate under the action of GCLC (glutamate–cysteine ligase catalytic subunit) [155]. If cystine input is blocked, intracellular GSH synthesis will be reduced, resulting in a decrease in GPX4 activity and ferroptosis.

Ferroptosis can also be induced by direct regulation of GSH or GPX4 decline. GPX4 can be regulated by inhibitors such as RSL3, ML162, ML210, and FIN56, while GSH synthesis is inhibited by butylthionine sulphoamine (BSO) [166]. The key factor of ferroptosis, Fe(II), can be inhibited in many ways, such as various inhibitors like deferoxamine (DFO), deferoxamine mesylate (DFOM), 2,2′-bipyridine (BP), and ciclopirox (CPX) [167]. Or it can be stored in the form of Fe(III) by ferritin and exported to the extracellular space via prominin2 (prom2) in the form of exosomes [168]. Ferroptosis can also be induced by promoting ROS production because of ROS’s strong oxidation. The deactivation of toxic lipid peroxides can also affect the occurrence of ferroptosis. The lipid antidote iPLA2β can turn PUFA-PL-OOH into non-reactive PUFA-OOH [169]. The process of ferroptosis, with its inhibiting factors and promoting factors, is illustrated in Figure 4.

#### 4.1.2. Ferroptosis-Related Genes Dysregulation and Therapy in BC

Tumor-infiltrating neutrophils (TINs), which inhibit tumor progression, are called immunosuppressive TINs [170,171]. In BC, immunosuppressive TINs can up-regulate aconitate decarboxylase 1 (*Acod1*) and promote the production of itaconate (ITA). ITA can activate *Nrf2*-dependent antioxidants, inhibiting lipid ROS production, thereby allowing tumor cells to evade ferroptosis and promoting lung metastasis of BC cells [170]. *Acod1* is a potential target for therapy to prevent tumor metastasis, and inhibition of *Acod1* may promote the ferroptosis sensitivity of immunosuppressive TINs and reduce tumor metastasis. Acyl-CoA synthase long-chain member 4 (Acsl4) is highly expressed in ER-negative BC cells and may support cancer cell growth [172]. In TNBC cells, oleic acid secreted by mammary fat cells can inhibit lipid peroxidation and then inhibit ferroptosis [173]. Acsl4 can use ATP to esterify CoA into free fatty acids [174]. Acsl4 can affect the occurrence of ferroptosis by influencing the synthesis of long-chain polyunsaturated fatty acids [174]. If the esterification reaction involved in Acsl4 can be guided in a direction conducive to ferroptosis, it may be conducive to promoting the death of ER-negative BC cells. Moreover, GPX4 was up-regulated in the luminal androgen receptor (LAR) subtype of TNBC. The inhibition of GPX4 can induce ferroptosis and enhance tumor immunity in LAR. The combination of GPX4 inhibitors with αPD-1 immunotherapy improves the efficacy of immunotherapy [175].

Other signaling pathways associated with ferroptosis can help BC therapy as well. Ferroptosis depends on the transport of iron ions and the inhibition of cystine import. Inhibiting ferroportin-1 (FPN) can disrupt iron transport, promoting the ferroptosis of BC cells [161]. In HER2-positive BC, fibroblast growth factor receptor 4 (FGFR4) accelerates cystine uptake and Fe^2+^ effect via the β-catenin/TCF4-SLC7A11/FPN1 axis and inhibits ferroptosis [176], indicating that it can be a potential target in BC. Additionally, glutathione cytoplasmic degrading enzyme (CHAC1) can enhance ferroptosis caused by cystine starvation through the GCN2-eIF2α-ATF4 pathway in TNBC [177]. In addition, hepatic leukemia factor (HLF), an oncoprotein in TNBC, regulates the secretion of growth factor 1 (TGF-β1) by tumor-associated macrophages (TAMs) by promoting IL-6. TGF-β1, in turn, promotes HLF to activate gamma-glutamyl transferase 1 (GGT1), which promotes drug resistance in tumor cells and TNBC cell proliferation and metastasis by promoting ferroptosis [178].

### 4.2. Cuproptosis

#### 4.2.1. Cuproptosis: A PCD Induced by Cu(I)

Cuproptosis was first defined in 2022 [15]. This type of cell death differs from other PCDs in its ultimate lethal substance. It is caused by excessive accumulation of copper ions (Cu^+^) in cells, leading to abnormal oligomerization and loss of activity of lipoylated proteins, and, finally, cell death [15,179,180]. Its toxicity comes from copper ions directly, while others, such as ferroptosis, stem from toxic lipid peroxides, or apoptosis from caspases.

Copper ions play a very important role in organisms. They are the metal center of many important enzymes in cell metabolism, such as CytC oxidase, superoxide dismutase (SOD), ceruloplasmin, lysine oxidase, etc. [181,182,183,184,185]. If the concentration of copper ions in the cell is too low, many metabolic enzymes will lose their function. Cu^+^ and Cu^2+^ exist simultaneously in the cytoplasm. As a reducing coenzyme factor, Cu^+^ can produce ROS through the Fenton reaction, like Fe^+^, if its concentration is too high in cells [181]. ROS can make mitochondria enter a stress state and cause damage, promoting mitochondrial autophagy [179,186,187]. In addition, copper ions, as the metal active center of SOD, can increase the enzymatic activity of SOD and scavenge ROS [185], while GSH can reduce toxicity due to its ability to bind Cu^+^ [188].

Cuproptosis depends on the key genes ferredoxin1 (FDX1) and thiocaprylsynthase (LIAS) [15]. The toxicity of copper ions mainly comes from the fact that cuprous ions can bind to the fatty acylated pyruvate dehydrogenase system, making them form oligomeric proteins, thus losing their activity [15,189] and blocking the formation of acetyl CoA. FDX1 can promote the fatty acylation of pyruvate dehydrogenase (PDH) and α-ketoglutarate, and can also reduce Cu^2+^ to Cu^+^, amplifying the toxicity of copper ions. LIAS can promote the synthesis of PDH cofactor lipoic acid (LA) and can help protein complete fatty acylation modification by the Fe-S cluster after binding with FADX1 [190], which is the final accomplice of cuproptosis. In this process, Fe-S clusters are consumed in large quantities, which destroys the electron transport chain [15]. At the same time, due to the blocking of pyruvate conversion to acetyl CoA, the tricarboxylic acid (TCA) cycle is inhibited, resulting in severe mitochondrial dysfunction and eventually leading to cell poisoning and death. The whole process of the cuproptosis pathway is illustrated in Figure 5.

Because the existence of copper ions may also cause autophagy and promote ferroptosis [179], the morphological characteristics of cuproptosis are not clearly defined at present. It is known that when cuproptosis occurs, due to the inhibition of the TCA cycle, biochemical indicators in cells will undergo significant changes, including but not limited to the accumulation of copper ions, pyruvate, and α-ketoglutarate, and the decrease in succinyl-CoA content [15,191]. These represent an important index to identify cuproptosis.

#### 4.2.2. Cuproptosis-Related Genes Dysregulation and Therapy in BC

Because of the proliferation of cancer cells, metabolic enzymes related to proliferation and respiration in cancer cells often have stronger activity than normal cells. As a coenzyme factor closely related to metabolism, copper ions have higher concentrations in cancer tissues than in normal cells [192]. Copper ions can promote angiogenesis [193], support cell respiration and metabolism, and assist gene transcription and expression [192,194,195]. Therefore, copper deficiency therapy [196,197] has been developed, which is a treatment method to suppress cancer by reducing the concentration of copper ions in the TME through the use of copper chelators. For TNBC stages III and IV, the copper chelator tetrathiomolybdate (TM) has demonstrated superior therapeutic efficacy and low toxicity [198].

Copper ions can promote PD-L1 expression on the tumor cell membrane surface [197,198], which is conducive to the localization of anticancer drugs targeting PD-L1. A higher copper ion concentration can promote the ubiquitination degradation of PD-L1 and promote the activation and proliferation of tumor antigen-specific CD8**^+^** T cells and NK cells [199]. In this case, copper-based nano-drugs have notable advantages. Nanogel prevents copper ions from directly entering the body for delivery and can be modified to regulate drug release and reduce drug toxicity. Lu et al. designed copper-based nanomedicines called CuETNPs that work against GSH to enhance the effects of copper ions, helping reverse cisplatin resistance and inhibiting tumor proliferation by activating cuproptosis [200]. NP@ESCu, a nano-drug synthesized by Guo et al. through a ROS-sensitive polymer and elesclomol copper ion, can assist αPD-L1 in cancer treatment by promoting PD-L1 expression on the tumor cell surface and improving the efficiency of immunotherapy [201].

Metal ion transport appears to be enriched towards TNBC tissue. If the levels of copper and zinc ions are increased in TNBC cells, the copper homeostasis of tumor cells will be destroyed, which can activate the cuproptosis of tumor cells, increase chemical sensitivity, and inhibit the proliferation of cancer cells [202]. Cuproptosis-related genes (CRGs) exhibit above-average expression in TNBC [202,203]. The copper transporter SLC31A1 has significantly higher expression in BC cells than that in normal cells [204,205]. Patients with slightly lower levels of SCL31A1 expression have a better prognosis [204] because the tumor cells absorb less copper ions for proliferation and angiogenesis, making the tumor cells easier to control. However, for patients with high SLC31A1 expression, this phenomenon can be exploited to design copper-loaded nano-drugs for treatment, because cancer cells absorb such drugs significantly more easily than normal cells, which can improve cancer treatment by activating cuproptosis. In addition, SLC31A1 is positively correlated with the expression of immune checkpoints CD274 and CTLA4/CD152, which are regulated by the LINC01614/miR-204-5p/SCL31A1 regulatory axis [205], suggesting SLC31A1 may be a potential target for adjuvant immunotherapy. Moreover, the cancer prognosis prediction of TNBC patients can be better performed to facilitate adjuvant cancer treatment by using CRGs [203].

## 5. Prospect

There are many methods of cell death, including autophagy, apoptosis, pyroptosis, necroptosis, ferroptosis, and cuproptosis, which have various regulatory signaling pathways and different mechanisms in different cancers and different drug treatments. In recent years, with the discovery of “PANoptosis”, which is the cross-regulation mechanism of PCD, parts of various death modes that have been researched separately in the past have more recently been linked together, suggesting that cell death regulation needs to be viewed more holistically.

### 5.1. Beyond PANoptosis, the Other Crosstalk of Cell Death in BC

The crosstalk between different cell death patterns occurs in various contexts, not only just between apoptosis, pyroptosis, and necroptosis, but also between ferroptosis, autophagy, and apoptosis. Autophagy mediated by ATG5 can degrade ferritin, including ferritin light polypeptide 1 (FTL1) and ferritin heavy polypeptide 1 (FTH1), and promote ferroptosis [206]. In BC cells, large amounts of iron-mediated ROS production can induce autophagy in the absence of ferroptosis [207]. In addition, the lncRNA LINC00618 can reduce the expression of SLC7A11, inhibit ferroptosis, and promote the expression of BAX, thus promoting apoptosis [208]. Erastin can trigger either ferroptosis or apoptosis, depending on the activation of PUMA. If PUMA induced by erastin is activated, it triggers apoptosis, while, if PUMA is inactivated, it triggers ferroptosis [209].

ROS has always occupied an important role in different cell death modes. ROS can promote the AMPK pathway and ATG4 oxidation to activate autophagy [210]. ROS can promote signaling pathways such as NF-κB and AP-1 [211], stimulate inflammation, promote MOMP, and release CytC [212] to promote apoptosis. ROS can activate pyroptosis via the ROS-JNK-caspase-3-GSDME axis [213]. Mitochondrial ROS can promote the autophosphorylation of RIP1, help RIP3 to be recruited into necrosome [59], and promote necroptosis. ROS also can promote ferroptosis through lipid oxidation [214]. In cuproptosis, copper ions can promote ROS production and induce oxidative stress, increasing the damage to cells [215]. ROS also can be inhibited by NFS1, which promotes Fe-S clusters, thereby affecting apoptosis, pyroptosis, necroptosis, and ferroptosis [216]. Cu(II) can promote ROS and consume GSH through the Fenton reaction, promoting ferroptosis [217]. In addition, the increase in Cu(II) can consume Fe-S clusters to promote cuproptosis [15]. The crosstalk between cell death can be better utilized in BC therapy, especially with the development of related drugs. In the future, research will definitely focus more on crosstalk in multiple cell death modes.

### 5.2. The Complexity of Cell Death in BC: Mutations and Drug Resistance

The mechanisms of cell death play a key role in the development of BC, and genetic mutations have a particularly significant impact on this progress. The PI3K/PTEN pathway, associated with autophagy, plays a vital role in BC cells. PI3K can promote the production of PIP3, thereby activating downstream effector molecules Akt/PKB. The activation of Akt not only affects autophagy but also regulates apoptosis [218,219]. PTEN is a phosphatase that dephosphorylates PIP3 to PIP2, serving as a negative regulator of the PI3K pathway, thus dampening the PI3K/Akt signaling cascade. Mutations or loss of function of PTEN have been identified in various cancers [220], leading to the sustained activation of the PI3K/Akt pathway, which promotes the proliferation, survival, and metastasis of tumor cells. Additionally, mutations occurred during the process of pyroptosis in BC cells. GSDME mutations at the caspase cleavage site and mutations at the N-terminal often occurred. These mutations significantly reduce spontaneous or drug-induced pyroptosis and promote the survival of BC cells [221].

In addition to genetic mutations, the death of BC cells is also influenced by other factor mutations, such as epidermal growth factor receptor (EGFR) mutations and ER mutations. When growth factors bind to EGFR, leading to the activation of multiple downstream signaling pathways, such as the MAPK/ERK pathway and the PI3K-AKT pathway, which are involved in regulating cell proliferation, differentiation, migration, and survival [222]. Overexpression or mutation of EGFR can lead to uncontrolled cell proliferation and survival, contributing to cancer progression [223]. In addition, ER mutations are also an important contributing factor in BC. ER mutations are closely related to endocrine therapy resistance in BC [224]. The discovery of these molecular mechanisms is critical for the development of new therapeutic strategies. It has been found that lapatinib can reduce the phosphorylation of EGFR and HER2, block Akt, MAPK, and mTOR signaling pathways, resist the inhibitory effect of insulin-like growth factor I (IGF-I) signal, promote apoptosis of BC cells, and reverse the therapy resistance of HER2-positive BC to trastuzumab [224]. In addition, lapatinib was shown to have only the adverse effects of conventional chemotherapy drugs, such as rash, diarrhea, nausea, and fatigue [225], and its combination with trastuzumab is not highly toxic.

Drug resistance has long been a topic in cancer treatment, and the drug resistance of cancer cells is often enhanced or weakened by the expression of certain genes under the chemotherapy drugs treatment, resulting in cell escape death. For example, the HR-positive BC patients receiving neoadjuvant chemotherapy, cancer cells often overexpress DNAJC12 to resist DOX through the DNAJC12-HSP70-Akt signaling axis, inhibiting ferroptosis and apoptosis, and causing drug resistance [226]. In addition, in ER-positive BC cells, after tamoxifen (Tam) treatment, transcriptional regulator nuclear protein 1 (NUPR1) expression was significantly up-regulated, and drug resistance was developed. This is due to the up-regulation of NUPR1, which causes the increase in autophagy, which in turn triggers the autophagy cytoprotection mechanism, and thereby cancer cells survive [227]. In TNBC, the overactivation of Src homology 2-containing phosphatase 2 (SHP2) leads to resistance to pyroptosis through JNK/NF-κB/Caspase-1/GSDMD pathway. Combining SHP2 inhibition with anti-PD-1 treatment has shown improved treatment efficacy [228].

It should be noted that the same BC therapy may lead to different modes of cell death in different BC cells. For example, cisplatin, a widely used chemotherapeutic drug, can induce various modes of cell death. In the MCF-7 (ER-positive) and MDA-MB-231 (triple-negative) BC cell lines, cisplatin can trigger autophagy and apoptosis [229]. However, the response of these two cells to cisplatin is not entirely the same. From the aspect of using cisplatin to induce pyroptosis, MDA-MB-231 cells are more prone to pyroptosis, while MCF-7 cells do not undergo pyroptosis due to cisplatin [230].

## 6. Conclusions

BC is the most common and most life-threatening cancer in women worldwide, with an increasing number of patients and deaths. The specific mechanisms behind the diverse subtypes of BC and the complex carcinogenic factors involved have not yet been fully understood. Because of chemotherapy resistance and the harmfulness of radiotherapy to the human body, the advantages of targeted therapy have been highlighted in cancer treatment, which generally leads to more effective treatment with fewer harmful effects. During the targeted therapy, identifying a target that can inhibit or eliminate cancer cells is crucial, not only for BC but for all types of cancer. Various studies have suggested the use of cell death as a target to inhibit cancer metastasis as an adjuvant treatment for BC. Taken together, the study of the death mechanisms of BC cells—autophagy, apoptosis, pyroptosis, necroptosis, ferroptosis, and cuproptosis—is of great significance for the treatment of BC, which not only provides many therapeutic targets for the development of drugs for the treatment of BC, but also provides more options for the clinical treatment of BC.

## Figures and Tables

**Figure 1 ijms-25-09703-f001:**
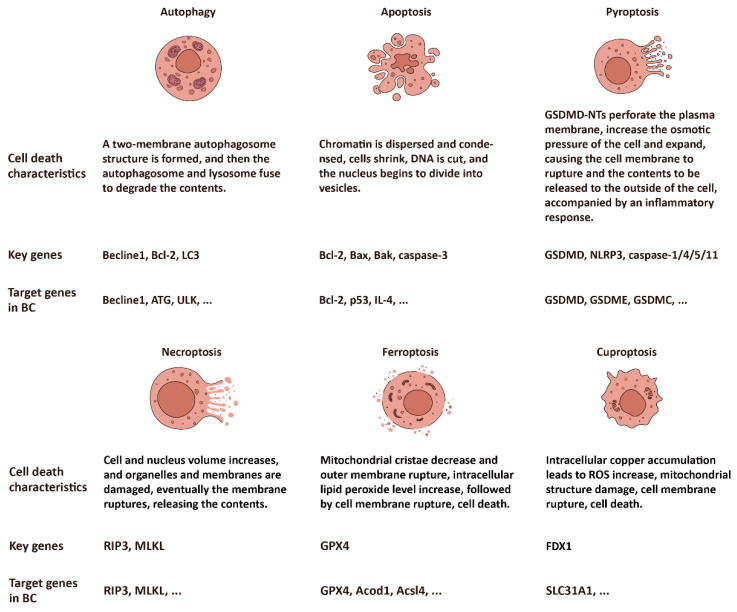
The profiles of six subtypes of cell death: autophagy, apoptosis, pyroptosis, necroptosis, ferroptosis, and cuproptosis.

**Figure 2 ijms-25-09703-f002:**
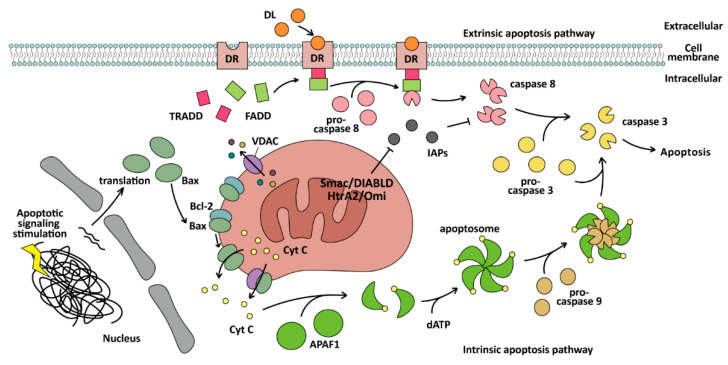
The mechanism of the intrinsic and extrinsic pathway of apoptosis. The intrinsic apoptosis pathway is mitochondrial-dependent and cleaves caspase-3 by caspase-9, while the extrinsic apoptosis pathway is mitochondrial-independent and cleaves caspase-3 by caspase-8, and exists in natural killer (NK) cells and CD8-positive cytotoxic T lymphocytes.

**Figure 3 ijms-25-09703-f003:**
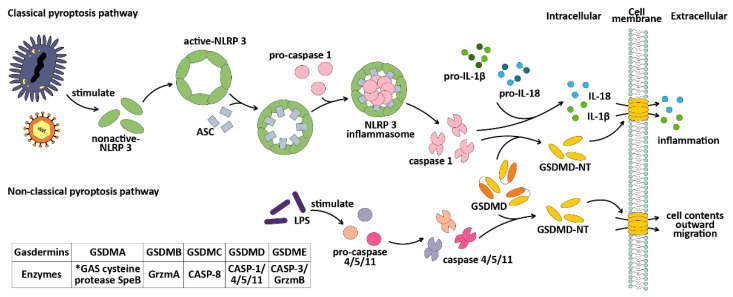
The mechanisms of the classical and non-classical pyroptosis pathways. Gasdermins can be cleaved by different enzymes, among which GSDMA can be cleaved by the GAS cysteine protease SpeB [122], GSDMB can be cleaved by granzyme A [123], GSDMC can be cleaved by caspase-8, GSDMD can be cleaved by caspase-1/4/5/11, and GSDME can be cleaved by caspase-3 or granzyme B. *GAS: group A *Streptococcus*. Grzm: granzyme. CASP: caspase.

**Figure 4 ijms-25-09703-f004:**
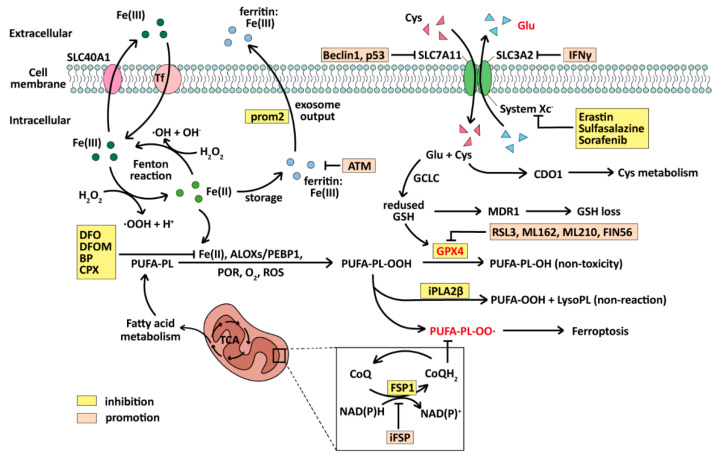
The mechanism of ferroptosis and its inhibitors. The whole process revolves around lipid peroxidation and depends primarily on Fe(II) involvement in the activation of the Fenton reaction and the inactivation of GPX4.

**Figure 5 ijms-25-09703-f005:**
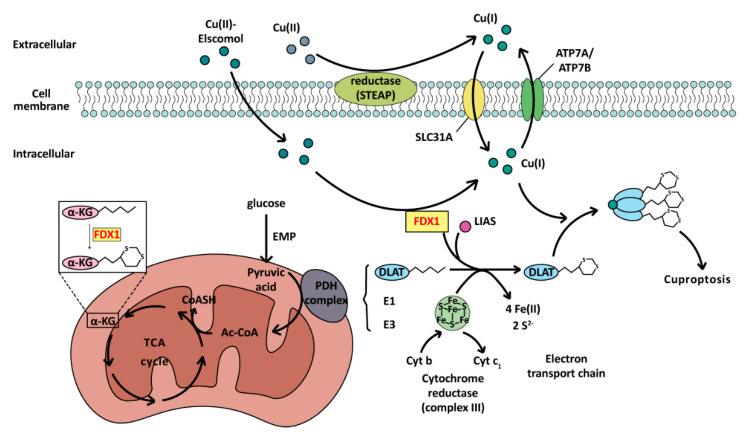
The mechanism of cuproptosis. As a cell death mode caused by exogenous factors, excessive accumulation of copper ions mainly affects the TCA cycle during cuproptosis. Dihydrolipoamide transacetylase (E2) of the PDH complex is oligomerized, regulated by ferredoxin FDX1 and the influence of copper ions, resulting in cell death.

## Data Availability

Not applicable.

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
