# Peer review of "Cell Death: Mechanisms and Potential Targets in Breast Cancer Therapy"

_ijms, 2024, doi:10.3390/ijms25179703_

Round 1
Reviewer 1 Report
Comments and Suggestions for Authors
The review paper provides insight into multiple programmed cell death, specifically in breast cancer cells, which are possibly employed in treating this type of cancer. Together with the use of traditional treatment including chemotherapy and radiation, supplements with adjuvant drugs targeting molecules involved in PCDs could further treat or inhibit the growth of breast cancer.
The idea is not so new, another better paper did write about PCDs and how targeting molecules in pathways of PCDs could either improve treatment sensitivity or inhibit cancer growth: https://www.nature.com/articles/s41392-022-01110-y. The significance of this paper is it more focuses on breast cancer and includes more PCDs but in less detail.
This review needs a significant revision before publishing. The qualities are significantly different across parts, maybe because more than one person drafted this paper and a throughout review was not there to keep the quality consistent.
The following revisions are suggested:
|
Line |
Comment |
|
50 |
What is the reason for the classification of Group A and Group B? |
|
69 |
KFERQ motif helps lysosomes to recognize and degrade proteins selectively |
|
70 |
in the reference you cite, target proteins are transported to the lysosomes by HSC70 and are recognized and internalized by LAMP2A |
|
74 |
What does it mean by “break them down into cells to provide energy” |
|
79 |
not only these four, but there are also ER-phagy, Lysophagy, Lipophagy... |
|
86 |
The sentence is awkwardly positioned, can just remove it, and what is full term of ATG? |
|
99 |
A poorly written sentences, makes it sound like autophagy receptors are encapsulated and degraded |
|
119 |
How do side effects and drug resistance relate to autophagy? The sentence is awkwardly positioned |
|
137 |
Another redundant and poorly positioned sentence. |
|
149 166 |
More like the change in miRNAs and lncRNAs expressions due to breast cancer affects autophagy than autophagy affecting BC |
|
181 |
the study cited here only says "inhibit the malignant progression of BC" not reverse. |
|
201 |
Reference |
|
217 |
BID not Bid |
|
231 |
not in NK cells. The way the author writes here sounds like extrinsic apoptosis only happens in NK cells. Besides NK cells, there is CD8-positive Cytotoxic T lymphocytes could also trigger extrinsic apoptosis |
|
286 |
English check |
|
321 |
PCD’s full term was already stated in the prior part. |
|
330 |
Nod-like receptors (NLRs) instead of NLRs (Nod-like receptors) |
|
400 |
Doxorubicin has been introduced in the past and now has a short-term |
|
425 |
A paragraph should have an overall sentence, and other sentences support that specific sentence. none of author sentences in this paragraph relate to others. Sentence line 425 is completely opposite and unrelated to others. Not only this paragraph, but most paragraphs in this paper also have similar problems, nonstructural, fragile, and containing unrelated sentences. |
|
423 |
Awkward short sentence without reference |
|
422 |
Which effects are similar to GSDMB? GSDMB as you write have both negative and positive effects to BC GSDMB-2 promotes tumor invasion GSDMB-3 and 4 induce pyroptosis, help slowing down BC other GSDMB isoforms inhibit pyroptosis??? |
|
445 |
Please be consistent in using full term and short term (BC) of Breast cancer |
|
487 490 |
No reference |
|
498 |
You mean PUFA-PL-OOH? |
|
515 516 |
These fundamental pieces information about ROS should have been mentioned earlier |
|
513 |
doesn't GPX4 inhibit Ferroptosis? |
|
520 |
Many pathways are not mentioned in the paper like iPLA2beta, fatty acid metabolism, DFO/DFOM/BP/CPX, Fe(III) transport, Fenton reaction... |
|
525 |
What is the other side of impact of TIN? |
|
531 |
What do you mean writing “promote the sensitivity of immunosuppressive TINs to ferroptosis” |
|
543 |
It is Luminal androgen receptor (LAR) |
|
556 |
how different? all PCDs have different mechanisms. Isn't it similar to ferroptosis as they are both involved in the accumulation of metal ions inside the cell? |
|
558 |
it is lipoacylated or lipoylated? |
|
667 |
please suggest molecule targets for inducing ferroptosis and cuprotosis in BC cells as you did for autophagy, apoptosis, necroptosis and pyroptosis |
Comments on the Quality of English Language
Mentioned above - inconsistent across parts
Author Response
The reply is submitted as attachment.

Reviewer 2 Report
Comments and Suggestions for Authors
Qian et al has submitted a manuscript entitled “Cell death: mechanisms and potential targets in breast cancer therapy” for consideration for publication. This review describes the different types of cell death and their role in breast cancer therapies. There have been many reviews in this area in the past couple of years and this review needs to be distinguish itself from others. The concept of PANoptosis is way to describe these different types of cell death in a new way. Unfortunately, this was not expanded beyond a description of the different types of cell death individually. List below are my concerns.
1. There are many similarities between apoptosis, pyroptosis and necroptosis. This was not emphasized. The role of caspases and how they can cross over between the different cell death mechanism as an example. Another example is reactive oxygen species and how it regulates these types of cell death.
2. Breast cancer therapies can induce different types of cell death. It would be helpful to describe how some breast cancer cells induce one type of cell death and other cells can induce another. The heterogeneity of cell death needs to be explored in more detail.
3. Genetic mutations and deletions were not discussed in enough detail. How does mutations in the PI3K/PTEN pathway affect cell death regulation? Beclin-1 haploinsufficiency was not mentioned.
4. Metal induced cell death is a great description of new types of cell death but again how does this overlap with other forms of cell death. For example ferroptosis could influence autophagy or apoptosis. The regulation of ROS is really important in these forms of cell death. How about heterogeneity of cell death following treatments that induce metal induced cell death.
5. Is their anything unique about breast cancer and how these cell death types are regulated? How about EGFR signaling or ER mutations?
6. How does drug resistance impact these types of cell death?
7. How will combination therapies improve breast cancer treatments without increasing normal tissue toxicity?
Comments on the Quality of English LanguageEnglish is fine.
Author Response
The reply is submitted as attachment.

Round 2
Reviewer 1 Report
Comments and Suggestions for Authors
No comments.